# Measuring Vegetation Heights and Their Seasonal Changes in the Western Namibian Savanna Using Spaceborne Lidars

Farid Atmani, Bodo Bookhagen * and Taylor Smith

Institut für Geowissenschaften, Universität Potsdam, 14476 Potsdam, Germany; atmanii.farid@gmail.com (F.A.); tasmith@uni-potsdam.de (T.S.)
* Correspondence: bodo.bookhagen@uni-potsdam.de

**Abstract:** The Ice, Cloud, and Land Elevation Satellite-2 (ICESat-2) with its land and vegetation height data product (ATL08), and Global Ecosystem Dynamics Investigation (GEDI) with its terrain elevation and height metrics data product (GEDI Level 2A) missions have great potential to globally map ground and canopy heights. Canopy height is a key factor in estimating above-ground biomass and its seasonal changes; these satellite missions can also improve estimated above-ground carbon stocks. This study presents a novel Sparse Vegetation Detection Algorithm (SVDA) which uses ICESat-2 (ATL03, geolocated photons) data to map tree and vegetation heights in a sparsely vegetated savanna ecosystem. The SVDA consists of three main steps: First, noise photons are filtered using the signal confidence flag from ATL03 data and local point statistics. Second, we classify ground photons based on photon height percentiles. Third, tree and grass photons are classified based on the number of neighbors. We validated tree heights with field measurements ($n = 55$), finding a root-mean-square error (RMSE) of 1.82 m using SVDA, GEDI Level 2A (Geolocated Elevation and Height Metrics product): 1.33 m, and ATL08: 5.59 m. Our results indicate that the SVDA is effective in identifying canopy photons in savanna ecosystems, where ATL08 performs poorly. We further identify seasonal vegetation height changes with an emphasis on vegetation below 3 m; widespread height changes in this class from two wet-dry cycles show maximum seasonal changes of 1 m, possibly related to seasonal grass-height differences. Our study shows the difficulties of vegetation measurements in savanna ecosystems but provides the first estimates of seasonal biomass changes.

**Keywords:** ICESat-2; GEDI; canopy height; lidar; savanna

## 1. Introduction

Above ground biomass (AGB) is a key variable for evaluating carbon sequestration and plays an important role in carbon cycle and climate change analyses [1]. Despite its importance, there is still considerable uncertainty in estimating carbon budgets [2], both locally and globally; an accurate, large-scale estimate of global AGB is thus a critical task for scientists. Canopy height is a key input for AGB estimation and plays a vital role in our understanding of the structure of vegetated terrain [3,4]. The use of field measurements to estimate canopy height is not feasible at regional or global scales, as it is often too expensive and time-consuming. Active remote sensing methods such as Light Detection and Ranging (lidar) and Synthetic Aperture Radar (SAR) combined with optical remote sensing data [5–12] have shown their effectiveness in measuring vegetation height and estimating AGB at the regional scale. Airborne and terrestrial lidar are widely used as a reliable technique to accurately map canopy height over small-to-medium scales (10 s of km$^2$) [13–15]. However, both terrestrial and airborne lidar data fail to estimate the canopy height at continental-to-global scales due to the limited spatial coverage and high acquisition costs. Mapping canopy height at larger spatial scales can be best achieved by spaceborne missions.

Two spaceborne missions were launched by the National Aeronautics and Space Administration (NASA) at the end of 2018: the Global Ecosystem Dynamics Investigation (GEDI) mission and the Ice, Cloud, and Land Elevation Satellite-2 (ICESat-2) mission. GEDI is a full-waveform lidar instrument onboard the International Space Station (ISS) launched on 5 December 2018 and has been collecting scientific data since April 2019 for a nominal two-year mission, which was recently extended until 2023 [16]. GEDI's three lasers operate at 1064 nm in the Near InfraRed (NIR) range of the electromagnetic spectrum with two full power lasers, one of which is split into two weaker energy beams (coverage laser), resulting in a total of four GEDI beams with a footprint of approximately 25 m in diameter and is thus significantly larger than airborne lidar footprints [17]. The four beams are optically dithered to produce eight ground tracks spaced approximately 600 m in the cross-track direction, resulting in a 4.2 km wide swath with footprints separated by 60 m along-track [18,19]. GEDI science data cover the Earth's surface between 51.6°N and 51.6°S [18].

ICESat-2's Advanced Topographic Laser Altimeter System (ATLAS) was launched in September 2018 to focus on the cryosphere, with global canopy height mapping as a secondary objective [20]. ATLAS operates at 532 nm in the green range of the electromagnetic spectrum, with a high sampling frequency of 0.7 m in the along-track direction and a footprint of approximately 13 m in diameter [15,16], and a temporal resolution of 91 days. ATLAS uses a multi-beam micro-pulse laser (photon counting), which makes it different from other lidar sensors in that the detectors are very sensitive to single photons. This means that any individual returned photon—whether from the reflected signal or solar background—can trigger an event within the detector; this results in a very noisy signal during daytime acquisition [21], making the identification of ground and canopy photons difficult [22]. Another limitation of ATLAS is that it may lose ground signal under dense forest or cloud cover due to the low energy of the laser. Sparse canopy cases also pose a challenge to vegetation height retrievals as canopy photons may be incorrectly identified as solar background noise. Therefore, the development of unique and effective algorithms is critical to differentiating signal photons from noise photons in diverse vegetation conditions.

The ICESat-2 science team has developed the DRAGANN (Differential, Regressive, and Gaussian Adaptive Nearest Neighbor) algorithm [23] to filter out noise photons from the ATL03 data product (geolocated photons) and classify the signal photons into ground and canopy photons with a 100 m step size in the along-track direction. The DRAGANN algorithm filters noise using an iterative gaussian filter based on the histogram of the nearest neighbors of each photon, where the ten largest Gaussian components within the histogram are removed iteratively. DRAGANN operates under the assumption that signal photons are clustered together, and noise photons are more sparsely distributed. Thus, canopy photons may be incorrectly identified as solar background noise in areas of sparse vegetation. Areas with canopy cover less than 15% are assigned as noise photons and will be filtered out by the DRAGANN algorithm [23]. The performance of the DRAGANN algorithm has been tested on simulated ATLAS data sets; results show that performance depends on several factors such as canopy cover, topography, and acquisition time [21]. Neuenschwander et al. [24] presented the first assessment of ICESat-2 ATL08 (canopy height) data over Finland and showed a Root Mean Square Error (RMSE) of 0.85 m for ground elevation and RMSE of 3.69 m for canopy height in a dense conifer forest area.

Several algorithms have been developed for filtering out noise photons from ICESat-2 data and to identify ground and canopy photons. Zhu et al. [22] proposed a localized statistics-based noise removal algorithm using simulated ICESat-2 data: Multiple Altimeter Beam Experimental Lidar (MABEL) [25]. The performance of the algorithm depended on the acquisition time (day- or night-time), canopy cover, and terrain slope. Wang et al. [26] verified the performance of the ground elevation and vegetation height retrieval algorithm [22] in retrieving ground photons in Alaska, finding that slope, signal-to-noise ratios (SNR), vegetation height, and vegetation cover all influence the ground classification. Their results showed an RMSE value of 1.96 m between the ground eleva-

tions retrieved from the ATL03 data and the corresponding airborne lidar data and that the ground elevation error is most affected by terrain slope. Popescu et al. [27] proposed an adaptive ground and canopy height retrieval algorithm based on photon clusters using the simulated ICESat-2 (MABEL) data [25] in different vegetation zones; the performance of the algorithm is less efficient in densely vegetated conditions. Zhu et al. [28] presented a modified version of the Ordering Points to Identify the Clustering Structure (OPTICS) algorithm, where the circular shape of the search area in the OPTICS algorithm is modified to an elliptical shape. The algorithm was tested using only the strong beams data in the Saihanba National Nature Reserve (pine forest and grasslands in Hebei, China), using airborne lidar data as a reference. A RMSE of 3.45 m for canopy height was found compared to a RMSE of 4.17 m using the original OPTICS algorithm. However, none of the previously proposed algorithms performed well in sparsely vegetated areas due to misclassification of sparse canopy photons as noise, where the DRAGANN algorithm overestimated the vegetation height in dwarf shrublands [29].

In this study, we estimated ground and canopy heights in the Namibian savanna using ICESat-2 and GEDI Level 2A ground elevation and relative height metrics product. Our study aims are: (i) to adapt ICESat-2 data to better classify sparse vegetation; (ii) to compare and validate tree-height estimates between spaceborne lidar datasets (GEDI, ICESat-2 ATL08) and field measurements; and (iii) to identify seasonal changes in vegetation cover. We improved the detection of sparse vegetation cover typical in savanna ecosystems by developing an improved noise filter and classification of ground and canopy photons from the geolocated data product ATL03 of ICESat-2. Our novel algorithm, the Sparse Vegetation Detection Algorithm (SVDA), reduces the influence of terrain slope in identifying and retrieving ground photons, as well as subsequent errors in defining canopy photons in sparsely vegetated areas. We emphasize that the presented algorithm for ICESat-2 is not a replacement for DRAGANN but is instead an alternative approach for savanna ecosystems that are characterized by sparse, isolated vegetation cover.

## 2. Study Area

The study area encompasses the savanna ecosystems located in the Omusati region in Northwestern Namibia, near the Etosha Pan (~$-19.45$ to $-18.77°$N, and ~$14.59$ to $15.94°$E) (Figure 1A, red polygon), covering more than 8000 km$^2$. Northern Namibia is characterized by an arid climate with two seasons: a rainy season from October to April (peak rain: February) with temperatures ranging from 20 °C to 34 °C during the day and a dry season from May to September with temperatures ranging from 18 °C to 25 °C during the day; mean annual precipitation is ~350 mm [30]. The terrain elevation of the study area ranges from 1090 to 1490 m (Figure 1A) and terrain slope ranges from 0° to 11°. The vegetation of the study area is dominated by tree and shrub savanna, with a Succulent Karoo biome characterized by shrubs and dwarf shrubs and a Nama Karoo biome characterized by deciduous shrubs and perennial grasslands [30].

We used two different geographic areas in our analysis: first, a larger area including low and moderate slopes and higher elevation, more densely vegetated terrain in the southern part of the study area (red polygon in Figures 1B,C and 2). This allows us to better validate the densely vegetated terrain. The second area is confined to the low slope savanna ecosystem with sparse vegetation (blue polygon in Figures 1B,C and 2).

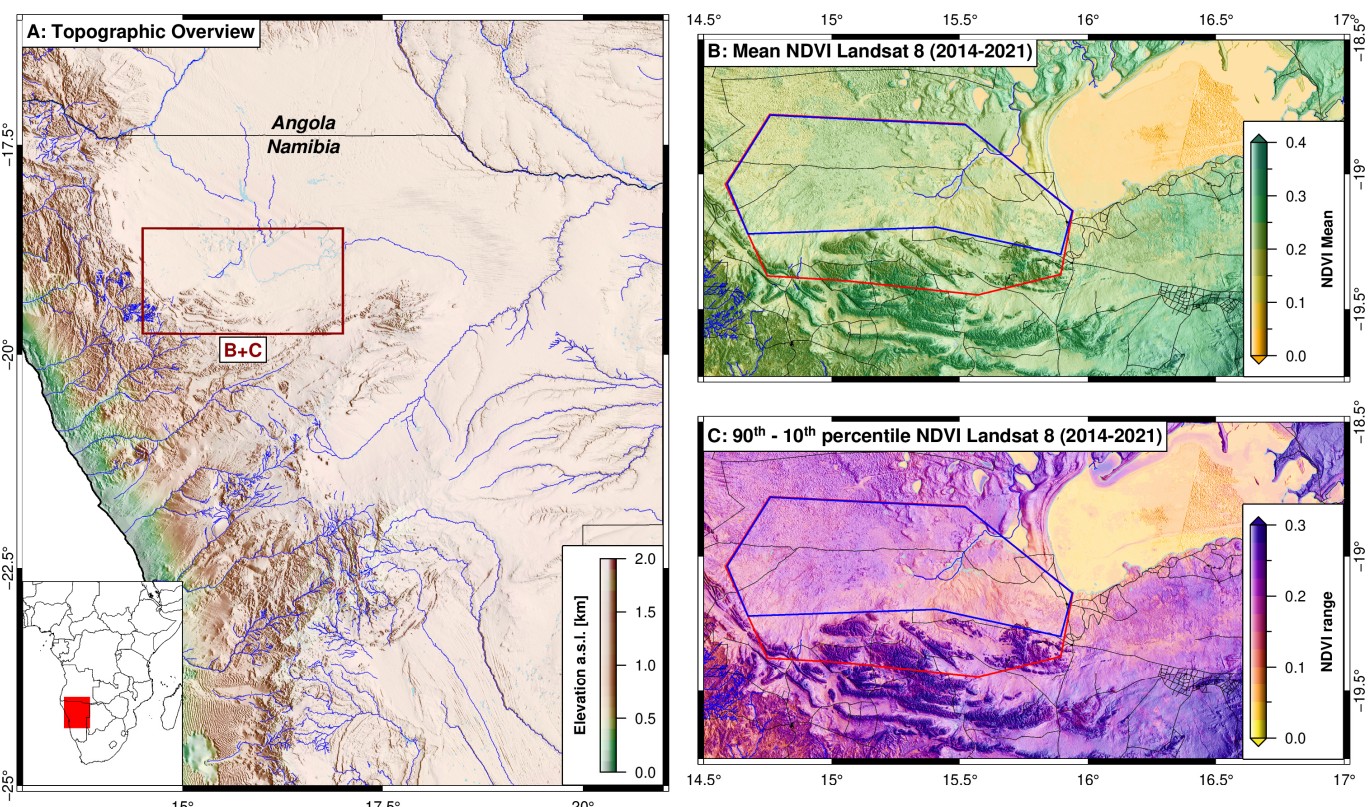

**Figure 1.** Geographic overview of the study area (**A**). International borders are indicated by black lines, lakes and wetlands are shown in light blue colors, and major rivers are in blue color. (**B**): Mean NDVI derived from Landsat 8 (2014–2021) for the study area including savanna and mountainous terrain (red polygon) and an area confined to the savanna ecosystem (blue polygon) [31]. Major roads are shown by black lines. (**C**): Range of the NDVI values between the 10th and 90th percentile showing the variability of vegetation cover. Higher values indicate higher seasonal vegetation changes.

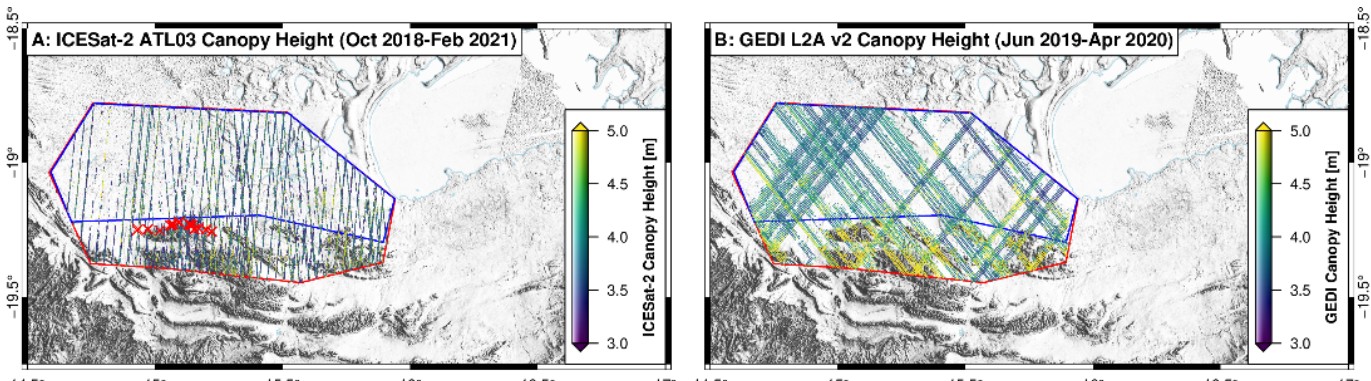

**Figure 2.** ICESat-2 and GEDI ground tracks for the study region (cf. Figure 1). (**A**) Canopy height derived from the SVDA processed ATL03 data described in this study. Red crosses indicate the 55 field-based tree height measurements of different species for validation. A detailed map of the tree-height field measurements is shown in Supplementary Figure S2. Red polygon outlines study area including more densely vegetated mountainous terrain and blue polygon is limited to the low-slope, sparsely vegetated savanna ecosystem. (**B**) Canopy height measurements taken from GEDI product L2A (version 2).

## 3. Materials and Methods

### 3.1. ICESat-2 Data

ICESat-2 ATL03 data product provides basic information on photons (latitude, longitude, and height referenced to the WGS84 geoid [32]) as well as a photon signal confidence level associated with each photon event. This serves as the input for each of the higher-level data products of Level-3 [21], which include the surface-specific data set ATL08 for vegetation and land elevations derived from ATL03 by the DRAGANN algorithm [21,24,32]. ATL03 and ATL08 data from 14 October 2018 to 14 December 2020 were downloaded from the NASA's National Snow and Ice Data Center (https://nsidc.org/, accessed on 21 September 2021) website using the boundary of the study area (Figures 1B and 2A).

### 3.2. GEDI Level 2A Data

The GEDI Level 2A data product with ground elevation and relative height metrics were downloaded from NASA's Earth Data website (https://search.earthdata.nasa.gov/, accessed on 28 December 2021) using the boundary of the study area. GEDI L2A data version 02 from 30 April 2019 to 02 February 2021 were downloaded (Figure 2B). Version2 GEDI data have a horizontal geolocation accuracy of less than 11 m and vertical accuracy of 10 cm [17].

The GEDI L2A algorithm [19] enables the derivation of footprint-level GEDI ground elevation and height metrics. Longitudes, latitudes, ground elevations, canopy heights, quality flags, degrade flags, and sensitivities were retrieved from each of the eight ground tracks. We limited our analysis to the high-quality returns: a quality flag value of 1 and a sensitivity higher than 0.95.

### 3.3. Sentinel Data

Sentinel-1 provides dual-polarization C-band Synthetic Aperture Radar (SAR) data [33], which we pre-processed using Google Earth Engine (GEE) [31]. The backscatter images in vertical–vertical (VV) and vertical–horizontal (VH) polarizations in the interferometric wide swath mode intersecting the study area from October 2018 to April 2021 were selected and a speckle filtering was performed with the focal median filter. For 156 scenes, we used the VV and VH polarizations to form a VV/VH ratio and calculated the median over the study period (October 2018–April 2021); we then extracted these data at each ICESAT-2 and GEDI footprint (Figure 3A). The VV/VH ratio indicates the depolarization of the radar signal; variations in this index have been previously used to identify forested vegetation and individual trees [34–36].

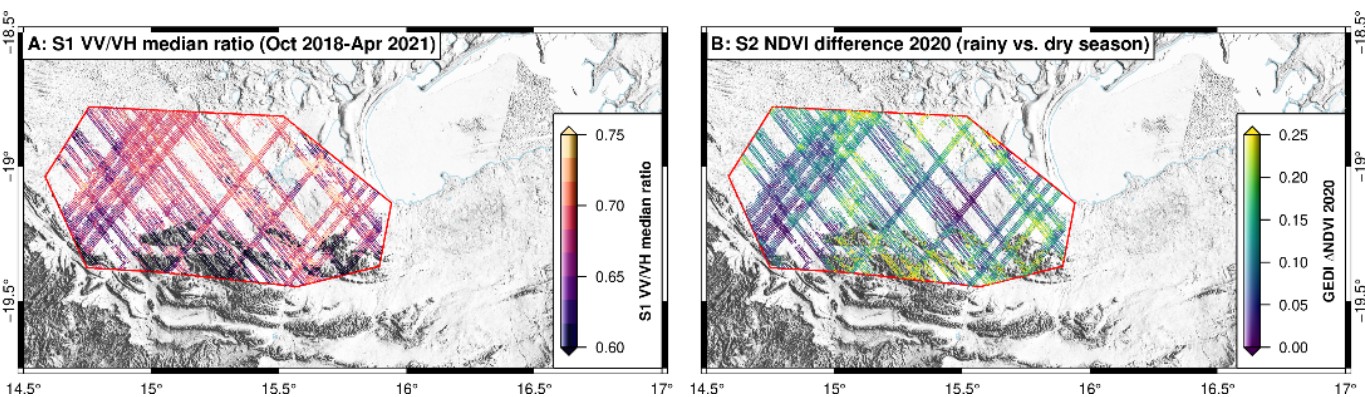

**Figure 3.** (**A**) Sentinel 1 median polarization ratio (VV/VH) averaged from October 2018 to April 2021 extracted for ATL03 and GEDI locations (only GEDI locations are shown here). Polarization ratios indicate the amount of depolarization usually associated with scattering on vegetation, especially trees. (**B**) Sentinel 2 NDVI differences between the rainy and dry season of 2020. The full S1 polarization ratio (VV/VH) coverage is shown in Figure S1.

Cloud-free optical orthoimages (Sentinel-2 Level-1C showing Top of Atmosphere reflectances) were downloaded from the United States Geological Survey website (https://earthexplorer.usgs.gov/, accessed on 13 March 2021). Scene-specific and seasonal-average (times: January/February 2020, June/September 2020) Normalized Difference Vegetation Index (NDVI) values were calculated; we also derived a seasonal NDVI difference between the rainy and dry seasons (Figure 3B).

### 3.4. Copernicus DEM

The Copernicus Digital Elevation Model (DEM) with a spatial resolution of 30 m was downloaded from the Copernicus Space Component Data Access system website (https://spacedata.copernicus.eu/, accessed on 9 June 2021). The DEM is derived from the data acquired during the TanDEM-X Mission; the horizontal reference datum is the WGS84 while the vertical reference datum is the Earth Gravitational Model 2008 (EGM2008) geoid. Elevations were transformed from the EGM2008 to the WGS84 datum before calculating the terrain slope. We used the Copernicus DEM, because this has been shown to have the highest precision among the global 30 m DEMs and the lowest inter-pixel noise (e.g., [37]).

The Copernicus DEM was used to validate the ground height from the three datasets, the elevation at the ATL03 SVDA ground photons, ATL08, and GEDI locations were retrieved from Copernicus DEM and compared to the measured ground height from ATL03 SVDA, ATL08, and GEDI. The validation of ground height was performed throughout the entire study and in the areas with low terrain slope (terrain slope < 5 degrees).

### 3.5. Field Vegetation Height Measurements

In situ tree height data were collected using distance and angle measurements during May 2021 for a total of 55 vegetation height measurements (Figure 2A and Figure S2 in Supplementary Material). Several tree species were measured in the field; samples are dominated by 39 Colophospermum Mopane, 11 Terminalia prunioides (family Combretaceae), 3 Dichrostachys cinerea, 1 Vachellia Reficiens, and 1 Commiphora glandulosa trees. The field measurements were compared to the top of the canopy of GEDI, ATL08, and the ATL03 SVDA.

### 3.6. Sparse Vegetation Detection Algorithm (SVDA)

Our proposed algorithm includes three steps for estimating ground elevation and canopy height using photon-counting lidar data, specifically ATL03 data: (1) retrieving the initial signal photons; (2) filtering out the noise photons using a local statistics approach; and (3) classifying the filtered photons data into ground, canopy, and top of canopy photons to extract vegetation height. An overview of the major steps is shown in a flowchart (Figure 4). We have implemented the SVDA in Python and provide the source code (see Data Availability Statement).

#### 3.6.1. Ground Photons Classification

In the first step of ground photons extraction, the ATL03 signal photons were divided into bins of 30 m in the along-track direction and the height was detrended in each bin; a threshold of $\pm 30 \times \tan(\pi/180 \times 30)$ was applied to the detrended signal to remove outliers; the threshold is calculated based on the maximum slope of the study area (30°) and the bin size (30 m) to preserve topography (see Figure S3 in Supplementary Materials). Next, an approach using percentiles was used to extract the ground photons and remove the remaining noise photons as follows: (1) the 25th and 75th percentiles of the photon height of each bin were calculated; (2) photons above the 25th and below the 75th percentiles of the photon height of each bin were extracted as preliminary ground photons (Figure 5A); (3) bins with more than 5 photons were selected; and (4) height was detrended a second time from the preliminary ground photons with a threshold of $\pm \tan(\pi/180) \times 30$ to filter the remaining noise photons. Bins with more than 5 photons were selected to calculate the final ground height. Finally, the median photon height of each preliminary ground photon

bin was calculated and assigned to the center of the bin and classified as the final ground photons (Figure 5B).

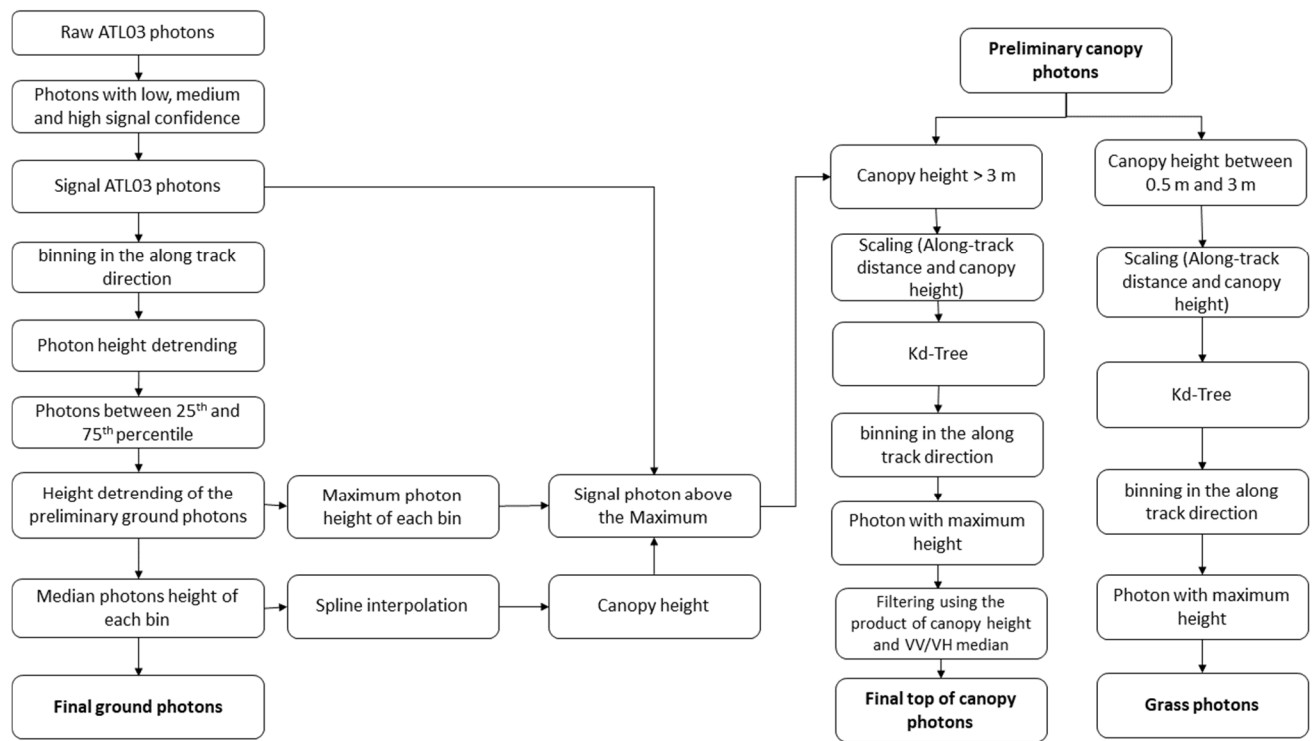

**Figure 4.** Flowchart of the Sparse Vegetation Detection Algorithm (SVDA) using ICESat-2 ATL03 data.

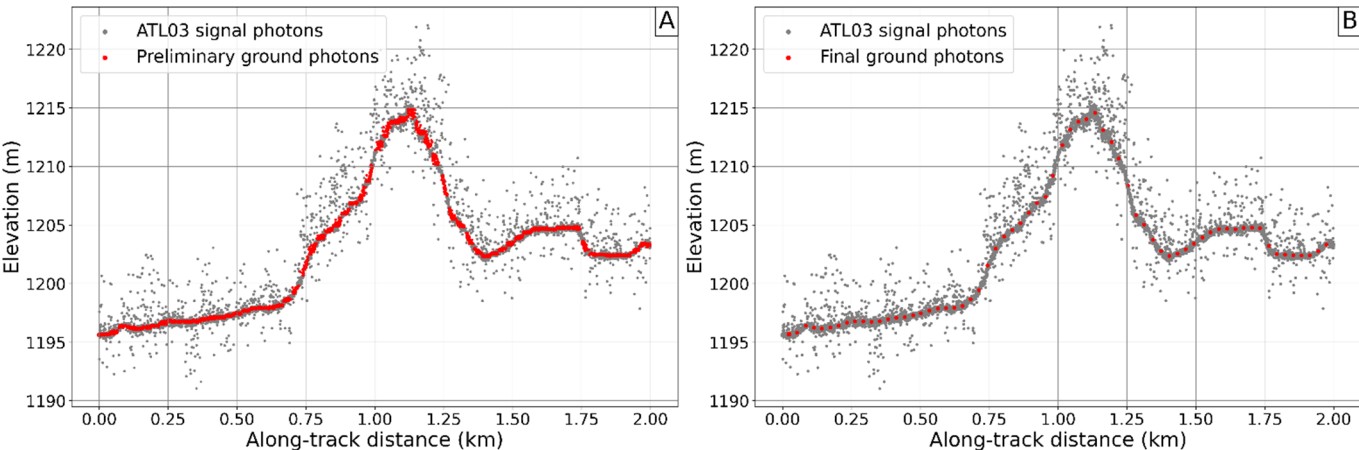

**Figure 5.** Characteristic example of ground-photon classification with gray dots showing all extracted signal photons from the ATL03 product (strong beam, gt1l, daytime acquisition with ID: ATL03_20190116100224_02890214_004_01). (**A**) Preliminary ground photons in 30 m steps are in red after extracting photons within the 25–75th height percentiles. This corresponds to step 1 and 2 as described in Section 3.6.1. (**B**) Final ground photons are selected based on additional filtering steps and detrending of height (steps 3 and 4 in Section 3.6.1).

### 3.6.2. Canopy Photon Classification

The canopy photon classification was performed using two main steps: The first step extracts the preliminary canopy photons based on the previously identified ground photons. The second step then filters the remaining noise photons from the preliminary canopy photons and classifies the photons into canopy and top-of-canopy photons.

The first step was performed in the following way: (1) the maximum photon elevation from each preliminary ground photon for all 30 m along-track bins (signal photons above the 25th and below the 75th percentile) was calculated. (2) All signal photons with an elevation above the calculated maximum were classified as preliminary canopy photons (Figure 6A). (3) A cubic spline interpolation was performed on the final ground photons and the canopy height was calculated for the preliminary canopy photons relative to the line.

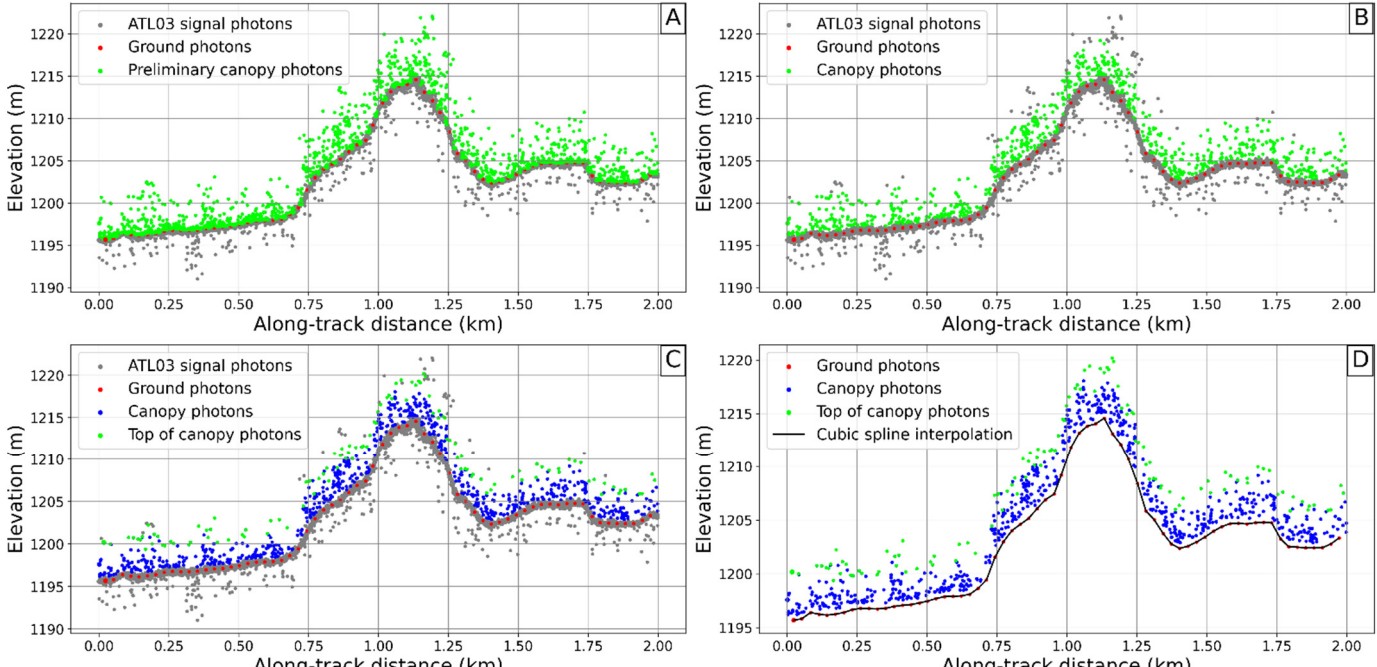

**Figure 6.** Characteristic filtering steps of a canopy classification exemplified on the strong beam, gt1l, daytime acquisition with ID: ATL03_20190116100224_02890214_004_01: (**A**) Ground photons in red derived from steps described in Section 3.6.1 and shown in Figure 5. (**B**) Preliminary canopy photons (green) as described in the first step in Section 3.6.2. (**C**) The second step described in Section 3.6.2 using height and neighbor filtering distinguishes between top of canopy (green), canopy (blue) photons, and the remaining noise (gray) photons. (**D**) Final photons classification with a cubic spline interpolation of the ground photons (black line).

The second step was performed as follows: (1) preliminary canopy photons with a canopy height equal to or above 3 m were extracted as trees, and photons with a canopy height between 0.5 m and 3 m were extracted. We acknowledge that grass heights do not reach 3m, but we wanted to cover the entire vegetation-height range. (2) The number of neighbors of each photon was calculated by linearly scaling the canopy height and along-track distance into values from 0 to 1 and the number of neighbors of each photon was calculated within a search area with a radius equal to 1% of the maximum Euclidian distance using a k-dimensional tree (Kd-Tree) for efficient neighborhood searches. (3) The preliminary canopy photons with a number of neighbors ≥ 6 were extracted as canopy photons (Figure 6B); the radius value and the minimum number of neighbors were selected by a trial-and-error approach but can be adjusted in the application of SVDA. (4) Canopy photons were divided into bins of 10 m in the along-track direction and the photon with the maximum canopy height among the neighbor-filtered (step 2) photons of each bin were classified as the top of canopy photon (Figure 6C,D). Finally, the top of canopy photons were filtered using the product of the minimum canopy height (3 m) and 80% of the maximum VV/VH median as a threshold. We used an averaged median VV/VH polarization ratio similar to previous studies to identify vegetation structure and height [34–36].

### 3.7. Canopy Height Comparison

Canopy heights were estimated from all data sets (ATL03 SVDA, ATL08, GEDI, and in situ data) and compared via height distributions and quantile-quantile (QQ) plots. A buffer of 10 m was applied to the ATL03 SVDA locations, and a buffer of 30 m was applied to the GEDI and ATL08 locations to consider the geolocation offset between the field and satellite data.

### 3.8. Seasonal Changes

The seasonal changes in tree height (canopy height ≥ 3 m) between the dry and the rainy season was performed by dividing the ATL03 SVDA, ATL08, and GEDI data into dry (May–October) and rainy seasons (November–April). The changes in vegetation with height between 0.5 m and 3 m between the rainy and the dry seasons was assessed using canopy height data from ATL03 SVDA. The datasets were split into rainy and dry seasons using the sensing date. Kernel density estimation was used to visualize the different percentiles of the rainy and the dry season data.

## 4. Results

### 4.1. ATL03 Signal Extraction

Filtering ATL03 photons with signal confidence is an efficient step in reducing the amount of noise photons and preserving the signal photons. Figure 7 shows the signal and noise photons from the strong and the weak beams at different acquisition times (day- and night-time acquisitions). The day-time acquisition (Figure 7A,C) data are characterized by low SNR due to the high sensitivity of the photon-counting lidar to individual solar photons; noise is much lower during night-time acquisitions (Figure 7B,D).

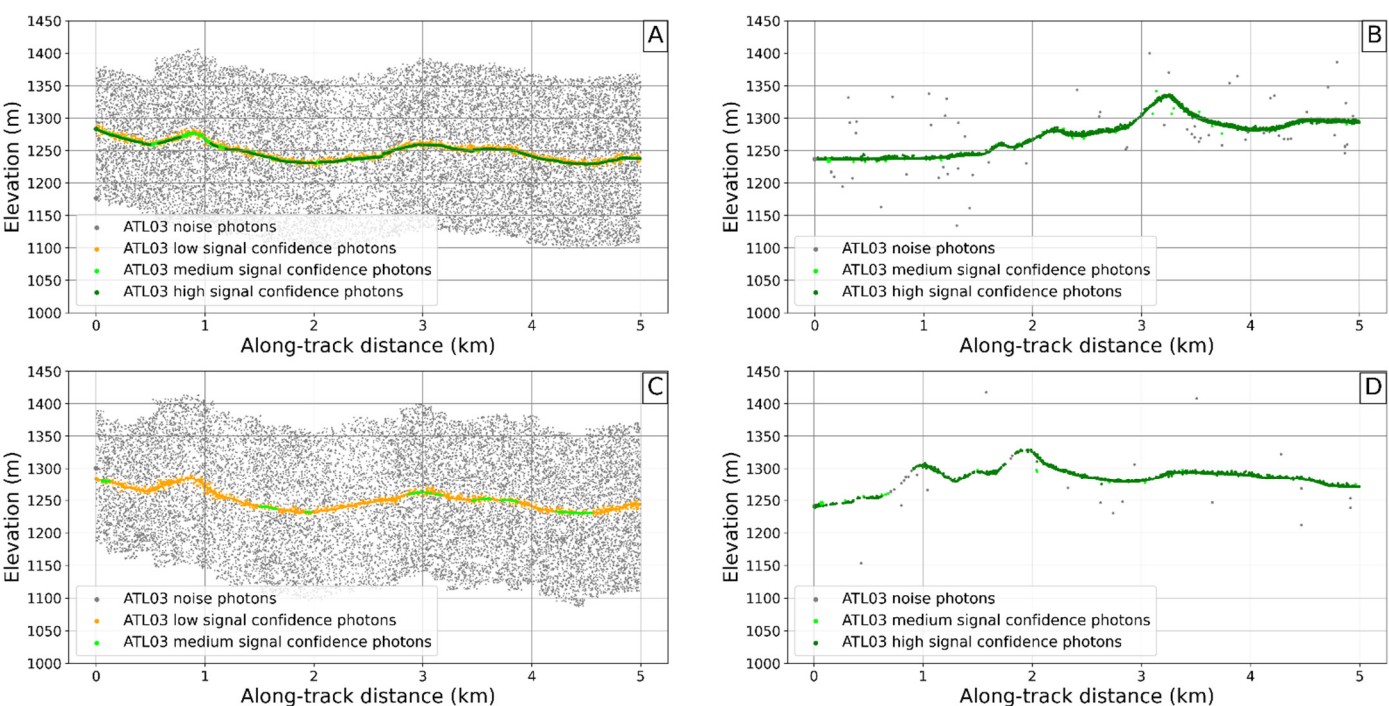

**Figure 7.** Signal photons extraction using the signal confidence flag for (**A**) strong beam during day-time acquisition, (**B**) strong beam during night-time acquisition, (**C**) weak beam during day-time acquisition, and (**D**) weak beam during night-time acquisition.

### 4.2. Ground-Height Validation

The ground heights derived from the ATL03 SVDA, ATL08, and GEDI were compared with Copernicus DEM elevation (Figure 8). The results of the ground height validation show that the ATL03 SVDA, ATL08, and GEDI ground height have RMSEs of 0.56 m, 1.51 m,

and 1.36 m, respectively. ATL03 SVDA performs notably better than ATL08 and GEDI ground heights; however, the number of comparison measurements is higher for ICESat-2 with 676,493 measurements from the ATL03 SVD, 37,633 measurements for ATL08, and 252,291 measurements from GEDI. We noted that higher slopes generally result in larger differences between spaceborne DEMs and Copernicus. When limiting our comparison to slopes below 5 degrees (see Figure S8 in the Supplementary Document), which is typical for the savanna environments in our study region, we observed a decrease in the RMSEs to 0.36 m, 1.03 m, and 0.85 m, respectively.

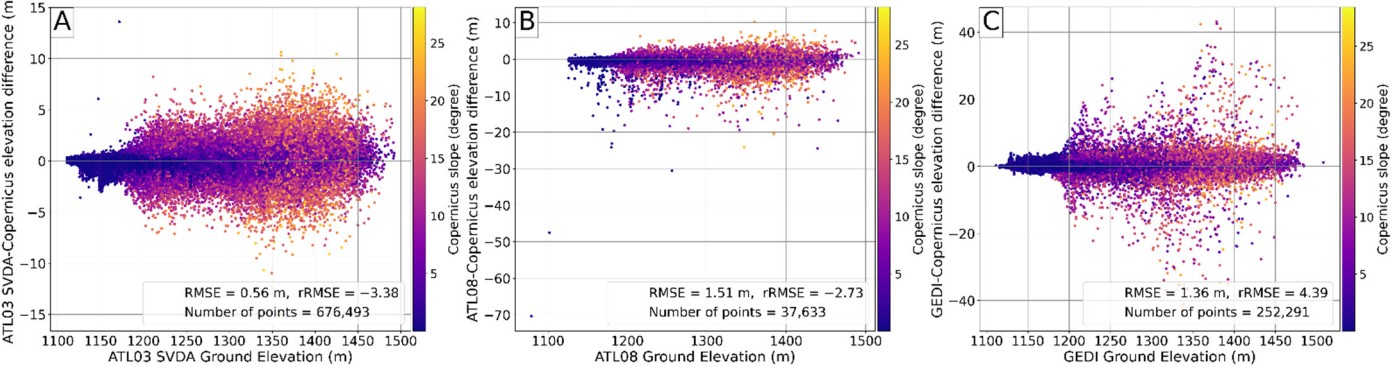

**Figure 8.** Copernicus DEM elevation and ATL03 SVDA based on ATL03 ground height measurements difference as described in Section 3.6.1 (**A**). (**B**): Copernicus DEM elevation and ATL08 ground height difference. We note that two ATL08 ground height points resulted in a difference more than 40 m compared to the ground elevation from Copernicus. (**C**): Copernicus DEM elevation and GEDI difference. rRMSE shows the relative RMSE (RMSE/mean of elevation difference). All elevation points are shown in this figure; see Supplementary Figure S8 for a comparison only of DEM slopes less than 5 degrees.

*4.3. Top of Canopy*

4.3.1. Top of Canopy Accuracy

The canopy heights derived from the ATL03 SVDA, ATL08, and GEDI were compared with field measurements (Figure 9). The results of the canopy height validation show that the ATL03 SVDA, ATL08, and GEDI canopy height have RMSEs of 1.82 m, 5.69 m, and 1.33 m, respectively. GEDI performs notably better than ICESat-2 (ATL03 SVDA and ATL08) canopy heights; however, the number of comparison measurements is higher for ICESat-2 with 39 measurements from the ATL03 SVDA and 22 measurements for ATL08. The distances between the intersected points are also different, as the canopy photons from the ATL03 SVDA intersect with the field measurements within a buffer of 10 m, while the intersections between ATL08 and GEDI with the field measurements are within a buffer of 30 m.

4.3.2. ICESat-2 ATL03 SVDA and ATL08 Comparison

Canopy photons of the two data sets that intersect were extracted and a canopy height comparison was performed over the entire study area and in a subset limited to the savanna ecosystem (red and blue polygons in Figures 1B,C and 2). The results show RMSEs of 3.7 m and 4.36 m for the entire study area (including mountainous terrain) (Figure 10A) and limited to the savanna ecosystem (Figure 10B), respectively. A total of $n$ = 37,701 intersecting points for the entire study area and a subset of 9076 for the savanna ecosystem were analyzed. There are a few outliers with a maximum difference of ~100 m caused by the failure of the ATL08 algorithm to accurately identify the canopy layer; there were no comparably large outliers in the SVDA data. In the entire study area, the distribution of the canopy height difference (Figure 10C) shows a mode of −0.88 m, mean value of −1.89 m, and a standard deviation of 2.26 m, indicating small canopy height difference between ATL03 SVDA and ATL08; the negative values of the mode and the mean indicate that ATL08

canopy heights are higher than ATL03 SVDA heights. In the savanna ecosystem data subset, the distribution of the canopy height differences (Figure 10C) shows a mode of −0.62 m, a mean value of −1.5 m, and a standard deviation of 2.25 m, indicating small canopy height differences between the ATL03 SVDA and ATL08; the negative values of the mode and the mean indicate that ATL08 canopy heights are higher than the ATL03 SVDA. Both differences in height distributions are confirmed by the canopy height distributions and the quantile-quantile plots (Figure S4), where ATL08 canopy height is higher at all quantiles.

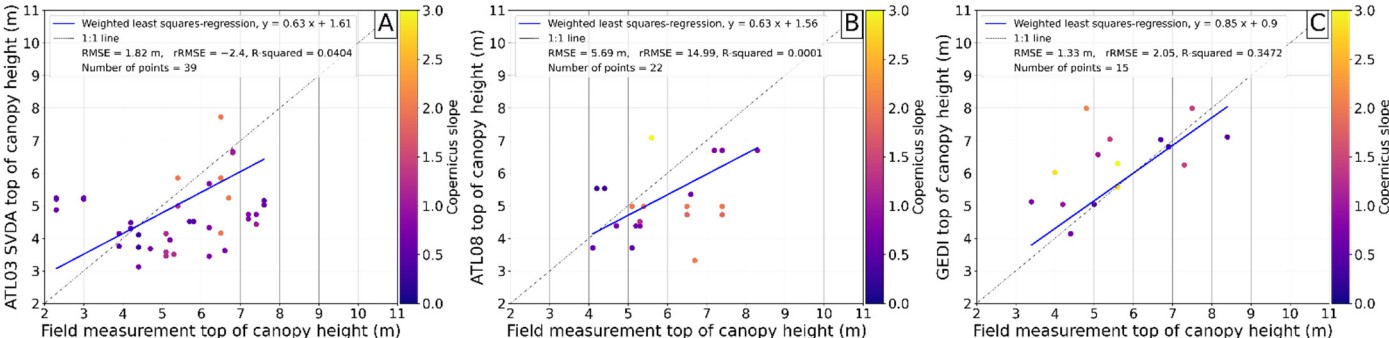

**Figure 9.** Field measurements (*n* = 55) intersection with ATL03 SVDA based on ATL03 canopy height measurements as described in Sections 3.6.1 and 3.6.2 within a buffer of 10 m (**A**). (**B**): Field measurements intersection with ATL08 within a buffer of 30 m. We note that one canopy height field measurements of 5.4 m resulted in an ATL08 canopy height of 31.24 m and we do not show this outlier to keep axes constant between the plots. (**C**): Field measurements intersection with GEDI within a buffer of 30 m. Blue line shows the weighted least squares regression, where the inverse of canopy height difference between GEDI/ICESat-2 (ATL03 SVDA and ATL08) and field measurements were used as weights. rRMSE shows the relative RMSE (RMSE/mean of the height differences between ATL and field measurements).

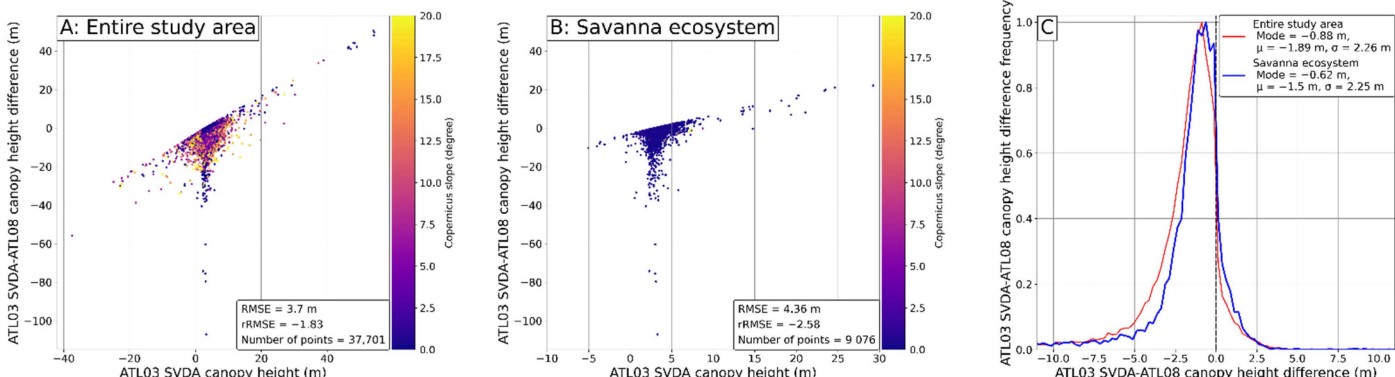

**Figure 10.** ATL03 SVDA and ATL08 canopy height relationship (**A**,**B**) for the study areas shown red and blue polygons in Figures 1B,C and 2. (**C**) The full distribution of canopy height differences, showing height differences of several meters in places and generally higher values of ATL08 estimates. rRMSE shows the relative RMSE (RMSE/mean of canopy height difference).

### 4.3.3. GEDI Level 2A and ICESat-2 SVDA Comparison

The canopy height relationship between the intersected returns of GEDI Level 2A and canopy photons from ATL03 SVDA within a buffer of 5 m (Figure 11A) shows 600 intersected points and a RMSE of 2.29 m. The geographic region encompasses the entire study area, as shown by the red polygon in Figures 1B,C and 2. The distributions of the canopy height differences of the intersected photons (Figure 11B) show a negatively skewed distribution with a mode of −0.25 m, mean of −0.94 m, and standard deviation of 1.76 m, indicating that the GEDI canopy heights of the intersected points are higher than

the ATL03 SVDA canopy height. The distribution shows that most of the intersections have a small canopy height difference of less than 2.5 m. The ATL03 SVDA and GEDI canopy height distributions of the intersected photons (Figure S5B) show that canopy heights from ATL03 SVDA are higher than GEDI between ~3 m and 3.8 m, and between 4.5 m and 5 m, while GEDI has higher canopy heights between 3.8 m and 4.5 m, and between 5 m and 12 m. GEDI and ATL03 SVDA have similar canopy height values at heights above 12 m.

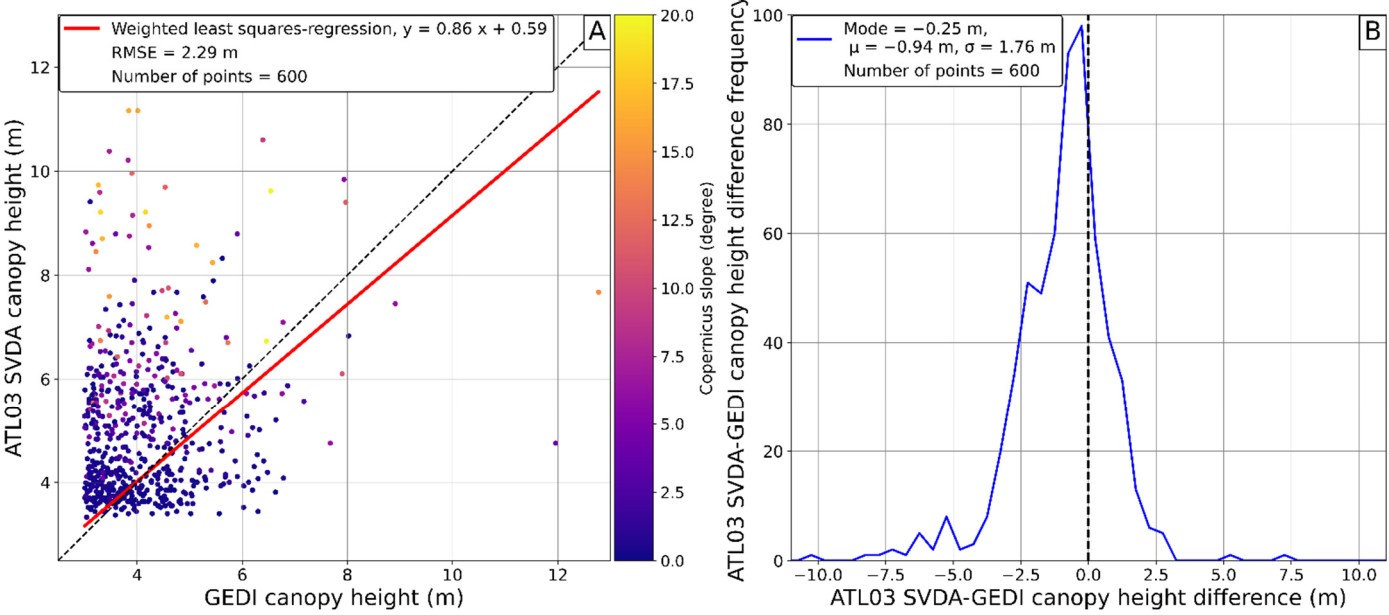

**Figure 11.** GEDI L2A version 2 and ATL03 SVDA canopy height relationship (**A**) and the distribution of ATL03 SVDA minus GEDI canopy heights (**B**). Comparison is based on 600 overlapping measurements within a buffer of 5 m and was carried out on the entire study area shown by the red polygon in Figures 1B,C and 2. GEDI canopy heights are generally lower than ATL03 SVDA canopy heights. Red line shows the weighted least squares regression, where the inverse of canopy height difference between ATL03 SVDA and GEDI measurements were used as weights.

Canopy height distributions of GEDI and the ATL03 SVDA of the entire study area (Figure S5C) and for the savanna ecosystem subset (Figure S5E) show that GEDI has higher canopy heights with values between ~3.5 m and ~4 m, while the ATL03 SVDA canopy height is higher than GEDI at values between 3 m and ~3.5 m and between ~4 m and ~8 m; the two datasets show similar canopy height values above 8 m in both areas. The distributions show mean values of 4.34 m and 4.17 m and standard deviation values of 1.07 m and 1.09 m for GEDI and ATL03 SVDA, respectively, in the entire study area. In the savanna ecosystem subset, the distributions show mean values of 4.07 m and 4.05 m and standard deviations of 0.73 m and 0.89 m for GEDI and ATL03 SVDA, respectively. The uncertainties are much lower in the savanna ecosystem subset compared to the entire study area. Quantile-quantile plots of ATL03 SVDA and GEDI canopy height distributions of the entire study area (Figure S5D) and in the savanna ecosystem (Figure S5F) show that ATL03 SVDA canopy height is higher than GEDI canopy height; similarity is observed at the very first quantiles.

### 4.3.4. GEDI L2A and ICESat-2 ATL08 Comparison

The canopy height relationship between the intersected returns of GEDI L2A and ATL08 canopy photons within a buffer of 5 m shows 91 intersected points and RMSE of 4.02 m (Figure 12A), with a maximum difference of ~25 m. The RMSE is strongly influenced by the outliers from ATL08.

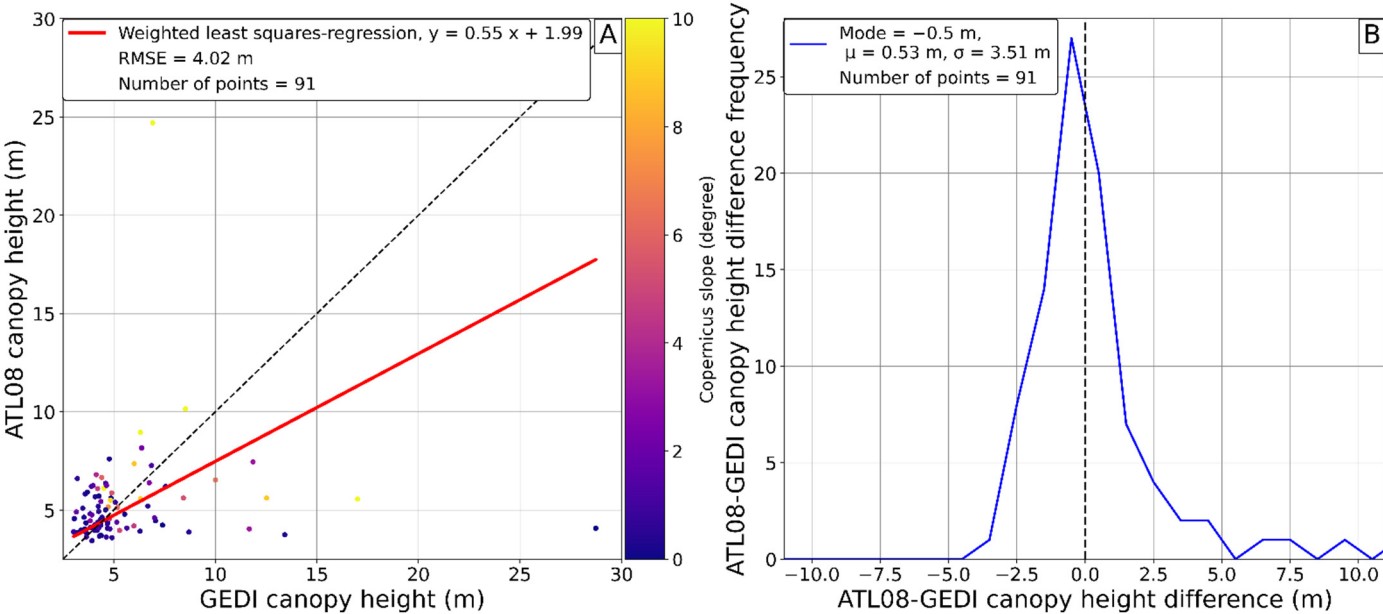

**Figure 12.** GEDI and ATL08 canopy height relationship (**A**) and the distribution of ATL08 minus GEDI canopy heights (**B**). Comparison is based on 91 overlapping measurements within a buffer of 5 m and was carried out on the entire study area shown by the red polygon in Figures 1B,C and 2. Red line shows the weighted least squares-regression, where the inverse of canopy height difference between ATL08 and GEDI measurements were used as weights.

The canopy height distribution of the intersection between GEDI and ATL08 within a buffer of 5 m (Figure 12B) shows a positively skewed distribution with a mode of −0.5 m, mean of 0.53 m, and a standard deviation of 3.51 m; most of the intersected photons have a canopy height difference less than 2.5 m. The ATL08 and GEDI canopy height distributions of the intersected photons (Figure S6B) show that ATL08 canopy heights are higher than GEDI between the canopy height values of 3 m and ~4.8 m, between ~7.8 and ~9.8 m, between ~10.5 m and ~14.2 m, and between 16.5 m and 18 m, while GEDI is higher between ~4.8 m and ~7.8 m. The two distributions show canopy height mean values of 5.06 m and 5.34 m and standard deviation values of 1.34 m and 2.44 m for GEDI and ATL08, respectively. ATL08 canopy height shows higher uncertainties caused by the misclassified photons.

ATL08 and GEDI canopy height distributions show that ATL08 canopy heights are higher than GEDI canopy heights for almost all the canopy height values. The ATL08 and GEDI canopy height distributions (Figure S6C) have means of 5.26 m and 4.28 m and standard deviations of 2.2 m and 1.14 m for the entire study area. In the savanna ecosystem subset (Figure S6E), the ATL08 and GEDI height distributions show mean values of 4.8 m and 3.99 m and standard deviations values of 2.08 m and 0.81 m. Quantile-quantile plots (Figure S6D,F) show that ATL08 canopy height is higher than GEDI canopy height at all quantiles.

*4.4. Seasonal Changes*

4.4.1. Tree Height

The seasonal changes in tree height (canopy height ≥ 3 m) between the dry and the rainy seasons (Figure 13) show an average increase of 39 cm from ATL08 and 19 cm from the ATL03 SVDA with different standard deviations for different datasets (σ ATL08 = 26 cm, σ ATL03 SVDA = 18 cm), while GEDI shows a decrease in tree height of 21 cm with a standard deviation difference of 6 cm. We have tested if the different height distributions are statistically different by applying a non-parametric Kolmogorov–Smirnov test. The canopy

height differences for vegetation heights above 3 m for the inter-seasonal comparison for ATL03 SVDA, ATL08, and GEDI data show *p* values below 0.05.

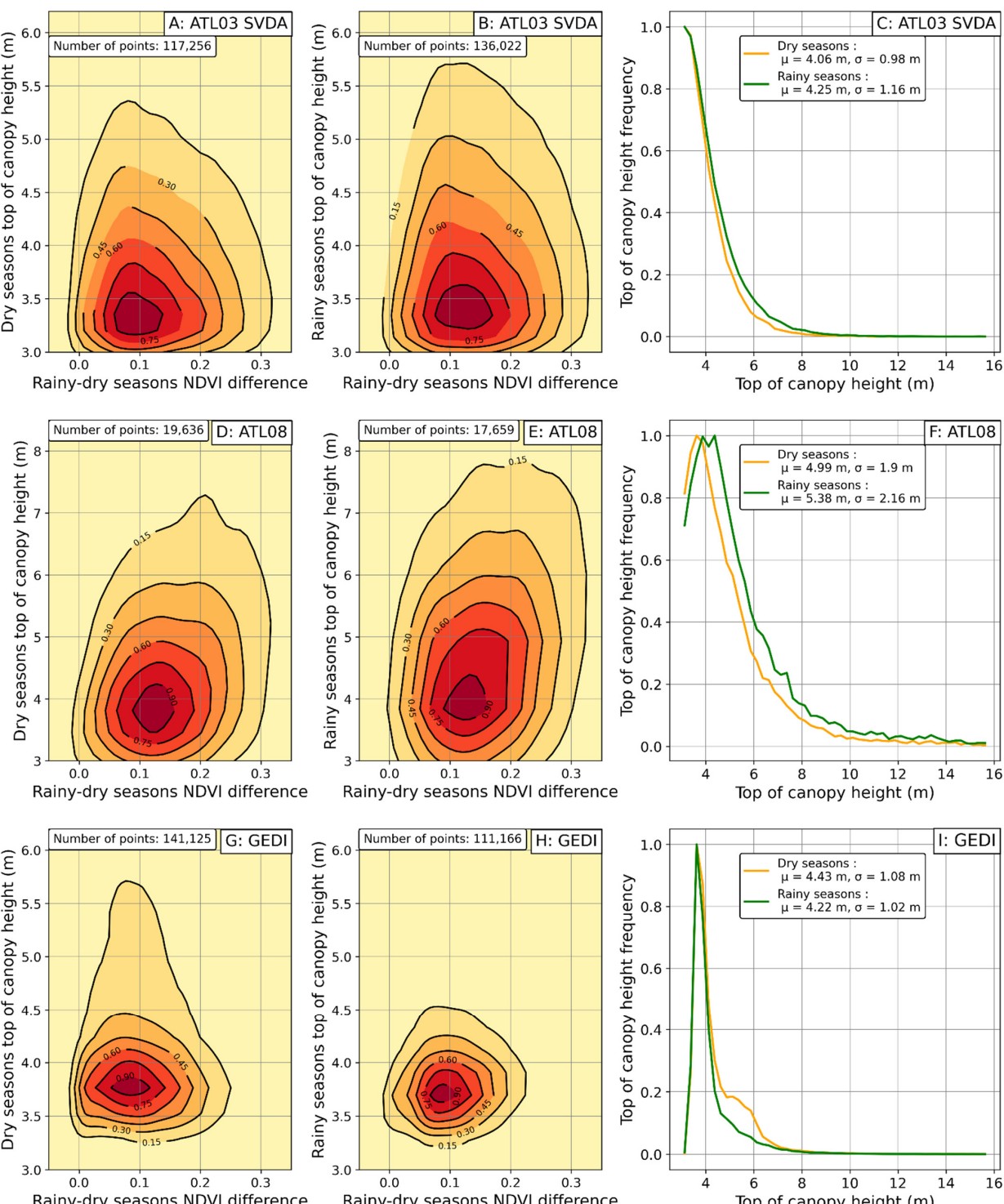

**Figure 13.** Top of canopy seasonal changes from ATL03 SVDA (**A–C**), ATL08 (**D–F**), and GEDI (**G–I**) data between the dry and the rainy seasons of 2019 and 2020. First and second column show the kernel density estimation (KDE) plots of the canopy height and NDVI difference distribution using a gaussian kernel with 100 equal steps for the x (NDVI difference from –0.05 to 0.35) and y axes (canopy height from 3 to 6.2 m for ATL03 SVDA and GEDI and canopy height from 3 to 8.5 m for ATL08) data. Left column: dry seasons, center column: rainy seasons. Third column shows the distributions indicating the small (but statistically significant) differences.

The maximum seasonal changes of tree height are different for the three datasets; kernel-density plots (Figure 13) show a maximum seasonal change of about 25 cm (70 cm) for the canopy height from ATL03 SVDA (ATL08). This positive trend in standard deviation demonstrates a seasonal increase in tree height (Figure 13C,F). Overall, the seasonal signal shows a consistent growth in tree height between the dry and the rainy seasons from the ICESat-2 datasets. Trees with heights between ~3.5 and ~9 m show the largest seasonal changes. The datasets show (Figure S7) different values of outliers (ATL03 SVDA = ~22 m, ATL08 = ~110 m, GEDI = ~26 m).

### 4.4.2. Vegetation Height Changes between 0.5 and 3 m

The seasonal vegetation height changes (0.5 m < canopy height < 3 m) between the dry and the rainy seasons of 2019 and 2020 show an average increase of 13 cm from ATL03 SVDA, with almost the same standard deviations between the rainy and dry season vegetation heights (standard deviation dry seasons = 0.65 m, rainy seasons = 0.69 m). We did not distinguish the vegetation type of vegetation heights below 3 m but note that grasses often dominate this area. A non-parametric KS test indicates that the dry and wet seasons are not drawn from the same distribution ($p = 0.0$). The seasonal vegetation height changes shown from the kernel density plots (Figure 14A,B) vary from ~10 cm to ~100 cm from the highest to lowest density. The seasonal signal (Figure 14C) shows consistent growth in vegetation height between the dry and the rainy seasons, which is expected from an area dominated by grasses.

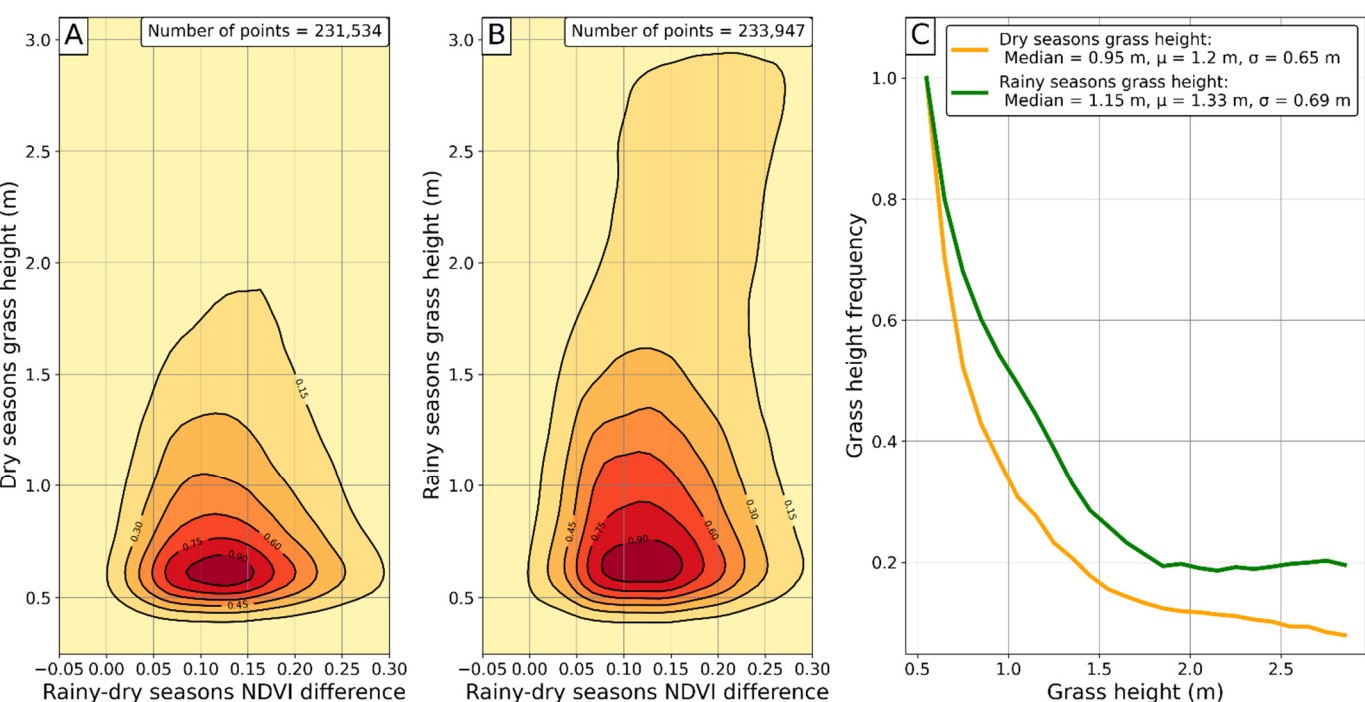

**Figure 14.** Seasonal vegetation height changes from ATL03 SVDA data between the dry and the rainy seasons of 2019 and 2020 for 231,534 (dry season) and 233,947 (wet season) points for the entire study area shown in the red polygon in Figures 1B,C and 2. (**A**) is the kernel density estimation of the joint distribution between NDVI difference and ATL03 SVDA dry season vegetation heights with a gaussian smoothing kernel and a step size of 100 in the x and y directions (with similar parameters in (**B**)). Left column: dry seasons, center column: rainy seasons. The differences are visible at all percentiles, but most strongly for the 30% of the data at higher vegetation heights. The distribution of the dry and rainy season vegetation heights (**C**) show the largest differences for higher vegetation height, which is expected in the seasonal savanna ecosystem. A non-parametric KS tests indicate that the seasonal vegetation heights are drawn from different distributions ($p = 0.0$).

## 5. Discussion

### 5.1. Ground-Height Measurements

In our study, we have evaluated ground and vegetation height measurements from different lidar data sets and present an algorithm for ground and vegetation heights measurement in savanna ecosystems with sparse vegetation cover. We show statistically significant canopy height differences between seasons, but emphasize the large uncertainties associated with space-born canopy height measurements.

The comparison of ground height from each dataset with the Copernicus DEM elevation has shown that ATL03 SVDA has the highest accuracy compared to ATL08 and GEDI. For ATL03 SVDA, the RMSE is below 1 m with $n = 676,493$, for ATL08 and GEDI, the RMSE is below 2 m with $n = 37,633$ for ATL08 and $n = 252,291$ for GEDI, indicating the high potential that these data have for measuring ground height. The results of the ground height validation show that the ATL03 SVDA, ATL08, and GEDI canopy height have RMSEs of 0.56 m, 1.51 m, and 1.36 m, respectively.

### 5.2. Vegetation Height Measurements: Caveats and Limitations

The comparison of canopy height from each dataset with the field measurements has shown that GEDI had the highest accuracy compared to ATL03 SVDA and ATL08.

However, the number of field measurements compared with each dataset is different, which complicates direct comparisons. For ATL03 SVDA and GEDI, the RMSE is below 2 m with $n = 39$ for ATL03 SVDA and $n = 15$ for GEDI, indicating the high potential that these data have for individual tree-height measurements. One of the factors that impacts the accuracy of individual tree height measurements is that lidar measurements are often not centered on trees compared to the field measurements where the maximum tree height is measured. This may lead to underestimated tree heights, which is clearly observed with canopy height from the photon-counting data (ATL03 SVDA in Figure 9A and ATL08 in Figure 9B). The underestimation of canopy height by ICESat-2 is also observed in forests and in overestimations of canopy height in dwarf shrublands [29]. Our results show some outliers from ATL08 with 77 photons (0.2% of all vegetation ATL08 photons) with canopy height values greater than 25 m, which can be considered invalid values. These outliers are caused by the failure of the DRAGANN algorithm to accurately identify canopy height; similar outlier problems are also observed in forested regions [38].

A direct comparison of vegetation heights without field measurements increases the sample size; the number of canopy height samples from GEDI and ATL03 SVDA are similar: ATL03 SVDA resulted in 255,578 samples and GEDI in 252,564 samples, while ATL08 returned 37,709 samples. Our data show that 81%, 83%, and 59% of canopy height values are lower or equal to 5 m and only 0.36%, 0.22%, and 4.87% of canopy height values are greater than 10 m in the study area for GEDI, ATL03 SVDA, and ATL08, respectively. The results of the canopy height accuracy assessment based on field measurements (Figure 9) show that 100%, 74%, and 77% of the intersected field measurements with GEDI, ATL03 SVDA, and ATL08 have canopy height values between 4 m and 8 m. The GEDI and ATL03 SVDA show a similar number of samples in this range, with 42.40% and 42.53% of all canopy height measurements falling in that range. Despite the similarity of GEDI and ATL03 SVDA at canopy height values between 4 m and 8 m, and canopy height distributions of GEDI and ATL03 SVDA (Figure S5), showing that ATL03 SVDA canopy height is higher than GEDI canopy height at the same range, ATL03 SVDA data underestimate, and GEDI data overestimate, canopy height. The differences in the datasets can be explained by different factors influencing canopy height classification such as terrain slope, the accuracy of terrain elevation classification from each dataset, and the location of the returned signal on the trees. We emphasize that despite these differences, the seasonal variation shows the same signal trends between the datasets, but with different magnitudes.

In addition to limited field-based canopy height measurements, we used a comparison of ground-classified points with the Copernicus DEM to identify consistency in height measurements. By relying on the low slope areas that characterize the savanna environments,

we observed a measurement error of 0.36 m for the ATL03 SDVA (*n* = 643,811). We carefully associated this value with a measurement uncertainty, although additional and precise field elevation measurements are required.

### 5.3. Vegetation Height and Polarized Ratio (VV/VH) Relationship

Canopy height and VV/VH median relationships (Figure S1 and Figure 15) show an expected negative linear correlation with a slope of −9.57, −7.62, and −5.16 for GEDI, ATL08, and ATL03 SVDA. The VV/VH median decreases with increasing canopy height, due to a larger increase in volume scattering in the cross-polarization (VH) than in the co-polarization (VV) direction. In trees with fewer branches and leaves, volume scattering is smaller than in trees with a higher number of branches and leaves. In this comparison, we do not distinguish between tree species, but suggest that the VV/VH ratio can provide additional information for species discrimination at large scales in savanna ecosystems. While the overall approach is promising, we caution that the VV/VH ratio is affected by changes in soil moisture and hillslope angle, and is impacted by tree location with respect to the Sentinel-1 pixel center.

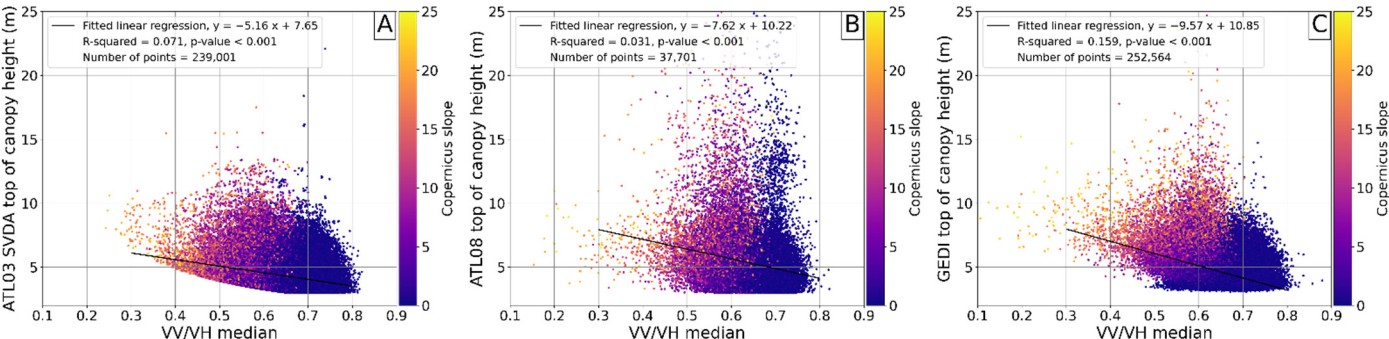

**Figure 15.** Canopy height from spaceborne lasers and Sentinel-1 VV/VH median relationships. (**A**) ATL03 SVDA canopy height and VV/VH median relationship, (**B**) ATL08 canopy height and VV/VH median relationship, and (**C**) GEDI canopy height and VV/VH median relationship. We note the weak, but statistically significant linear relation between VV/VH and canopy height measurements.

Due to significant radar ground-noise scattering, we note the inherent noise in the relation (Figure 15), but also emphasize the statistically significant relation. Our comparison only includes pixels with vegetation heights above 3 m, as identified by the spaceborne lidar data (GEDI and ICESat-2).

### 5.4. Seasonal Vegetation Height Changes

ATL03 SVDA and ATL08 show consistent seasonal changes in tree height with an average change of 19 cm ± 18 cm and 39 cm ± 26 cm. The standard deviations are propagated from the standard deviation difference between the rainy and dry seasons and are larger than the height difference. The impact of leaf coverage is stronger during the rainy season as compared to the dry season and, thus, tree-height is higher during the rainy season. However, the impact of vegetation structure is different between the two types of lidar systems and algorithmic differences may enhance these differences. GEDI shows a decrease in tree height during the rainy seasons, which can be explained by the low sensitivity of full waveform lidar to the leaf changes and their associated volumes.

We aimed to map seasonal changes in vegetation height, especially vegetation with heights of less than 3 m, because of the importance for above-ground biomass estimation in savanna ecosystems. For example, the grass layer plays an important role for the savanna biome for the food web, wildlife, and wildfire occurrence [39,40]. Savanna ecosystems are characterized by a continuous grass layer and discontinuous tree layer [41], and the grass layer is an important sink for above-ground carbon in savanna ecosystems. GEDI

and ATL08 cannot map grass height as the minimum canopy height from GEDI is 3 m and from ATL08 it is ~2 m. However, we argue that we successfully mapped vegetation height and its seasonal changes with the ATL03 SVDA algorithm, where the differences are observed at all percentiles, with a maximum height change of 1 m observed in 15% of the height data. The ATL03 SVDA can be used to map the lower vegetation layers in the savanna ecosystems and its seasonal changes, thus contributing to the efforts in accurately mapping above-ground biomass and reducing the uncertainties in estimating above-ground carbon stocks.

## 6. Conclusions

In this study, we compare different spaceborne laser systems for monitoring canopy height changes in a savanna ecosystem. The study was performed in northwestern Namibia near the Etosha Pan, where vegetation is characterized by continuous layers of grass and isolated trees. We relied on ICESat-2, GEDI, and Sentinel-1 radar polarization data. Because of the unique vegetation characteristics in savanna ecosystems with large continuous grass coverage and sparse individual trees, an accurate mapping of canopy height and its seasonal changes is challenging, as each lidar sensor has its limitations.

We present an algorithm for sparse vegetation detection in savanna ecosystems, which includes improved noise filtering, as well as ground and canopy photon classification using ICESat-2 ATL03 data. We refer to this algorithm tailored to the savanna ecosystem as the ATL03 SVDA (Sparse Vegetation Detection Algorithm). A qualitative evaluation of the performance of the developed algorithm, the ICESat-2 ATL08 data product, and the GEDI L2A data product for canopy height retrievals was performed, and seasonal changes in canopy were estimated. In total, 55 field measurements were used as reference canopy heights. The following main conclusions can be drawn:

First, GEDI L2A performed best in retrieving absolute tree height with a RMSE of 1.33 m compared to ATL03 SVDA with a RMSE of 1.82 m and ATL08 with RMSE of 5.69 m. The proposed ATL03 SVDA is effective and improved the identification of canopy height by utilizing percentile statistics and the number of neighbors of each photon. ATL03 SVDA provides a valuable basis for estimating low vegetation height and its seasonal changes using photon-counting lidar data.

Second, the photon-counting lidar (ICESat-2) shows a positive seasonal tree height change with an average change of 19 cm from ATL03 SVDA and 39 cm from ATL08. However, the full wave form lidar (GEDI) shows a negative seasonal change with an average change of 21 cm, which we explain by the low sensitivity of full waveform lidar to leaves.

Third, seasonal vegetation-height changes from ATL03 SVDA show a maximum vegetation height change of 1 m between the rainy and the dry seasons for parts of northern Namibia. We associate this vegetation height with seasonal grass height changes but emphasize that we do not distinguish between different vegetation types.

This study shows that ICESat-2 and GEDI hold great potential in estimating canopy height. Our ATL03 SVDA can effectively estimate and map tree and vegetation heights and their seasonal changes. The algorithm can help to overcome the limitations of the ATL08 algorithm in accurately retrieving canopy heights and provides additional information on the dynamic changes of vegetation in savanna ecosystems. Our results provide a valuable means of reducing the uncertainties in estimating above-ground biomass and carbon stocks in sparsely vegetated regions.

Continuing research will focus on the application of the ATL03 SVDA for larger areas and a data fusion between lidar, optical, and radar data to improve the detection of seasonal vegetation changes. We emphasize the need for data fusion between multi-spectral and high-temporal resolution optical and SAR data to improve vegetation classification. More refined knowledge of vegetation structure will allow improved lidar-classification approaches.

**Supplementary Materials:** The following are available online at https://www.mdpi.com/article/10.3390/rs14122928/s1, Figure S1: Sentinel 1 polarization ratios; Figure S2: ICESat-2 and GEDI ground tracks and tree-height measurement field sites; Figure S3: Filtering threshold explanation; Figure S4: ATL03 SVDA and ATL08 canopy height comparison; Figure S5: GEDI L2A and ATL03 SVDA canopy height relationship; Figure S6: GEDI L2A and ATL08 canopy height relationship; Figure S7: Seasonal tree height changes from ATL03 SVDA; Figure S8: Ground height validation with Copernicus DEM.

**Author Contributions:** Conceptualization, F.A. and B.B.; analysis, F.A., B.B. and T.S.; writing, all authors lead by F.A. All authors have read and agreed to the published version of the manuscript.

**Funding:** This research is part of the project "Options for sustainable land use adaptations in savanna systems: Chances and risks of emerging wildlife-based management strategies under regional and global change (ORYCS)" and is funded by the Federal Ministry of Education and Research (BMBF) within the funding measure SPACES-II (Science Partnerships for the Assessment of Complex Earth System Processes). Grant number FKZ 01LL1804A.

**Data Availability Statement:** IceSat-2 data can be obtained from NASA's National Snow and Ice Data Center (https://nsidc.org/, accessed on 21 September 2021) and GEDI data are available at NASA's Earth Data website (https://search.earthdata.nasa.gov/, accessed on 28 December 2021). The SVDA algorithm is available on github with processing examples described in several Jupyter Notebooks (https://github.com/UP-RS-ESP/ICESat-2_SVDA.git, accessed on 28 December 2021).

**Acknowledgments:** We acknowledge tree-height field measurements from Josef Haitula (NUST), Hesekiel Na-tangwe Akwenye (NUST), Robert Hering (UP), Helena Wiskott (UP), and Tim Herkenrath (UP). We thank Niels Blaum (UP) for discussion. Funding was provided to the ORYCS project by the B.M.B.F. We are grateful to the three reviewers who have provided valuable comments that improved the manuscript.

**Conflicts of Interest:** The authors declare no conflict of interest.

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
