# Peer review of "Measuring Vegetation Heights and Their Seasonal Changes in the Western Namibian Savanna Using Spaceborne Lidars"

_remotesensing, doi:10.3390/rs14122928_

Round 1

Reviewer 1 Report

This paper proposes a new algorithm for the calculation of canopy height using a combination of ICEsat-2 and sentinel-1 datasets for a location in Nambia with a dominant land cover of savanna grasslands and relatively sparse tree-cover. The authors compare the results of their method against two existing publically available products (for ICEsat-2 and GEDI) and validate the results against a small number for field validation plots. The manuscript is of interest for the scientific community, there are however several issues that need to be addressed before the manuscript can be considered for publication.

The manuscript would greatly benefit from restructuring and sections being expanded upon, particularly the methods section. I found it hard to follow and understand what the authors actually did and how. The method section needs to be edited to include all analyses, including pre-processing steps and what each dataset was used for and why. Two datasets are listed in the methods section and never mentioned again, here sentinel-2 imagery and a Copernicus elevation model product. Likewise, very few details are given in the methods about what field data was measured and how it was used for validation. The goals of the paper are also not clear - the comparison of the three elevation datasets is stated, but no mention of the assessment of seasonal changes is given until the results section (4.3).

I have provided many additional comments inside the PDF document.

Reviewer 2 Report

Vegetation canopy height is a key factor for estimating aboveground biomass. How to use spaceborne lidars to accurately obtain vegetation canopy height is a hot topic. This paper used two spaceborne lidar data to explore their ability to measure the vegetation height at a savanna ecosystem. The major contribution of this paper is proposed a new algorithm to filter the ICEsat-2 photons. However, this manuscript needs to be large improved before reaching this journal’s publish quality. Following are the detail comments.

  1. The description of this new method needs to improve. For example, what is the reason to use 25th and 75th height percentiles to select the preliminary ground photons? Why the median photon height can be used as the ground photons? Why used 3m to classify the grass photons and other canopy photons?
  2. Which canopy height product is used to filter the canopy photos? Moreover, what is standard to filter the top of canopy photons with VV/VH median?
  3. Does the ATL03 SVDA and ATL08 used the same track or used the same criteria to filter the data? Why they used the different buffer size to match the ground observation? Moreover, the influence of geolocation error needs to be further discussed.
  4. Have you split the ICSsat-2 data into different time or strong and weak beams while comparing canopy height among ATL03 SVDA, ATL08 and GEDI? Many studies show acquisition time and beam type have a large influence to accuracy of canopy height.
  5. What is the reason for decease in tree height between the dry and the rainy seasons in GEDI? Do the tracks of GEDI in dry or rainy seasons were different which could induce the bias in sample of vegetation? Moreover, the comparison in seasons change is conducted in the entire area or only the savanna ecosystem which should be clarified in the method part.
  6. The seasonal changes in vegetation heights need validation which would provide confident result for the reader. Otherwise, it is only presented the data comparison between different time, and we cannot know change in height is caused by the tree height change or the measurement uncertainty.

Some specifical comments:

  1. Line 27 the “Ground” should not be capitalized.
  2. Line 188 Which part used the Copernicus DEM ?
  3. Line 215 “±30*tan(π/180)*30” or “±tan(π/180)*30” ?
  4. Line 249 the typo of “photos”
  5. The section 4.1 is basic information of exacting ICEsat-2 photons which in unnecessary to be presented here.
  6. What does the contour line mean in the Figure 12 and 13?

Reviewer 3 Report

I reviewed your manuscript with interest. Below are my comments to consider for improving your work:

Major issues

1. The threshold used in the study need to be scientifically justified. Some of them are reported as arbitrary or lucky picks. Two examples are: 1) why a bin soze of 30 m?  I would expect choices to compare with e.g. footprint level lidar like 55m (ICESat-1), 25m (GEDI) etc. 2) Why between 25th and 75th percentile for ground photons? I would expect subsampling areas with known ground elevation and calibrate this threshold based on the percentile that match these known elevations. What makes one not choose e.g 33% to 66% (1/3 to 2/3)?

2. Report relative RMSEs (e.g as a percentage of the mean or median) – They are better to understand than absolute values

3. Methods for seasonal changes analysis are not adequately explained. Moreover, what you report are seasonal variations in each sensor’s capability to measure height, not seasonal growth. The trees in this ecosystem are deciduous so we expect parameters detected under leaf-on conditions to differ from those detected under leaf-off conditions. Your language sounds like the trees were changing heights, which is not true.

4. You claim that “We successfully mapped grass height” and yet again you admit that “We acknowledge that grass heights do not go up to 3 m, but we wanted to cover the entire vegetation-height range“. This is confusing and inconsistent. Moreover, in your study ecosystem, heights below 3 m comprise of grass and shrubs You are better off calling this 0 - 3m height category accordingly.

5. According to your conclusion – “First, GEDI L2A performed best in retrieving absolute tree height with a RMSE of 533 1.33 m compared to ATL03 SVDA with a RMSE of 1.82 m and ATL08 with RMSE of 5.69 but GEDI cannot map grass while your SVDA does”. I generally agree with this conclusion confirming that global algorithms normally don’t work for localized savanna studies, thus we need locally derived algorithms/models. The question now is how can these two approaches be fused for a better savanna ecosystem structural study?

Minor issues

1. Are you sure the rainy season is still October-April? Seasons in the Southern African region have generally shifted and the rains start mid -to-late November now.

2. Isn’t Acacia reficiens now Vachellia reficiens?

3. The use of adverbs doesn’t sound scientific without statistical back-up e.g line 273, very efficient?? Something like "..reduces noise photons by XX%.." would sound better

Round 2

Reviewer 1 Report

The authors propose a new method to remove noise and estimate (ground and canopy vegetation) height using Icesat-2 data for a location in Namibia with a dominant land cover of savanna grasslands and relatively sparse tree-cover. The results of which are compared with two existing satellite elevation products. In addition, the detection of seasonal temporal changes is presented. The manuscript is improved after the previous round of revision, and represents a new and novel approach. The expansion of information on data sources used was helpful. I do believe, however, the manuscript should be further improved in order to be ready for publication.

The stated objectives of the paper, with regard to the presented results, omit two key points: the evaluation of optimum time of data capture and the evaluation of seasonal changes. Currently only two objectives are stated: (1) the development of a new method for a savanna context, and (2) the comparison of height estimate accuracy generated from two existing height products (GEDI & ATL08) against the new SVDA.

The methods section would be improved by expanding the information given about the use of Sentinel-1 data. As it stands, there is one sentence concerning this. Some details about what was implemented is given in the discussion section. It appears that the approach was used to filter the data in some manner, and was therefore critical to the overall method in calculating top of canopy height. This needs better documenting.

Following on from the above, regarding the discussion section “5.3 Vegetation height and polarized ratio (VV/VH) relationship”: contains elements critical to methods section and also presents results. Moving elements of this section to the methods and results would make the decisions made over the course of the research more understandable.

I have provided additional comments inside the attached PDF document.

Reviewer 2 Report

The authors have addressed my concerns in the last reversion. I have no more questions about the manuscripts. However, there still a few suggestions for the authors.

1)    Figure 9. It would be more confident if the authors provide the R2 for each comparison?

2)    Figure 15. It would be better to use “ p-value < 0.001” than  “ p-value = 0.0 ” or “ p-value = 3.2e-256”
